# Incidence and predictors of antiretroviral treatment failure among children in public health facilities of Kolfe Keranyo Sub-City, Addis Ababa, Ethiopia: Institution-based retrospective cohort study

**Meseret Misasew**[1]*, **Takele Menna**[2], **Eyoel Berhan**[2], **Daniel Angassa**[3], **Yesunesh Teshome**[4]

**1** Center for Disease Control and Prevention, Addis Ababa, Ethiopia, **2** St. Paul's Hospital Millennium Medical College, Addis Ababa, Ethiopia, **3** MERQ Consultancy, Addis Ababa, Ethiopia, **4** Clinton Health Access Initiative, Addis Ababa, Ethiopia

* mesimisasew@gmail.com

**Data Availability Statement:** All relevant data are within the manuscript and its Supporting information files.

## Abstract

### Background

Human immunodeficiency virus (HIV) infection is a public health concern globally. The number of people living with HIV worldwide in 2018 was estimated at 37.9 million; of those, 1.7 million are children. Globally, 62% of the 37.9 million people were receiving Antiretroviral treatment (ART); and among those who were on ART, 53% had achieved viral suppression. This study aimed to assess the incidence and predictors of Antiretroviral treatment failure among children in Kolfe Keranyo sub-city, Addis Ababa, Ethiopia.

### Methods

An institution-based retrospective cohort study was conducted among 250 children who were enrolled in first-line Antiretroviral treatment from January 2013 to May 2020 in Kolfe Keranyo sub-city. Data was collected by using a data extraction checklist and data were extracted by reviewing children's medical charts and electronic database. Kaplan–Meier method was used to estimate the probability of treatment failure. During bivariable analysis variables with p-value < 0.25 were taken for multivariable Cox regression analysis to assess predictors of treatment failure. Statistically significant association was declared at p-value < 0.05 with a 95% confidence interval.

### Result

The overall proportion of treatment failure within the follow-up period was 17.2%. This study also found that the overall incidence rate was 3.45 (95% CI: 2.57–4.67) per 1000 person-month observation. Infant prophylaxis for PMTCT (AHR: 3.59, 95% CI: 1.65–7,82), drug substitution (AHR: 0.18, 95% CI: 0.09–0.37), AZT/3TC/NVP based regimen (AHR: 2.27,

**Funding:** The authors received no specific funding for this work.

**Competing interests:** The authors have declared that no competing interests exist.

**Abbreviations:** 3TC, Lamivudine; AIDS, Acquired Immune Deficiency Syndrome; ART, Antiretroviral Treatment; AZT, Zidovudine; DTG, Dolutegravir; HAART, Highly Active Antiretroviral Therapy; HIV, Human Immunodeficiency Virus; IRB, Institution Review Board; LPV/r, Lopinavir / Ritonavir; NNRTI, Non-Nucleotide Reverse Transcriptase Inhibitor; NRTI, Nucleotide Reverses Transcriptase Inhibitor; NVP, Nevirapine; OI, Opportunistic Infection; PI, Protease Inhibitor; PMTCT, Prevention of Mother to Child Transmission; SPSS, Statistical Package for Social Sciences; TB, Tuberculosis; VL, Viral Load; WHO, World Health Organization.

95% CI: 1.14–4.25), and more than 3 episodes of poor ART adherence (AHR: 2.27, 95% CI: 1.17–4.38) were found to be predictors of treatment failure among children.

## Conclusion

High proportion of treatment failure was found among children on first-line ART in Kolfe Keranyo sub-city, Addis Ababa according to the UNAIDs virological suppression targets. Infant prophylaxis for PMTCT, drug substitution, AZT/3TC/NVP based initial regimen, and poor ART adherence were found to be predictors of first-line ART treatment failure. Close follow-up of children on medication adherence and revising the AZT/3TC/NVP based regimen need to be considered.

## Introduction

Human immunodeficiency virus (HIV) infection is a public health concern globally, especially in Sub-Saharan African countries. The number of people living with HIV worldwide in 2018 has been estimated at 37.9 million out of which 1.7 million were children. Sub-Saharan Africa remains the region most heavily affected by HIV, accounting for 67.5% of HIV infections worldwide in adults and 90% in children [1–4]. New HIV infections in 2018 accounted for 1.7 million adults and 160,000 children and 54% of children living with HIV were receiving life-long ART in low and middle-income countries [5].

Ethiopia, to mitigate the impact of HIV, began providing Antiretroviral Treatment (ART) in 2003 and free ART was launched in 2005 [1, 6]. According to the 2017 Ministry of Health estimate, 722,248 Ethiopians were living with HIV but with a low ART coverage of 61% for adults and 33% for children, far from the UNAIDS target of 90% [7]. In Ethiopia, there is a significant pediatric HIV-1 burden with approximately 63,227 children living with HIV and an estimated 5479 new HIV-positive children with 3200 AIDS-related children deaths occurring annually [1, 2].

Monitoring of ART has been used to ensure successful treatment, identify adherence problems, and determine whether ART regimens should be switched in case of treatment failure [2, 3]. Treatment failure is a suboptimal response or a lack of sustained response to treatment [8, 9]. Treatment failure types are clinical, immunological, and virological failures.

The global target of a sustainable developmental goal (SDG) is to achieve 95-95-95 treatment targets at the end of 2020. These targets are first 90% of people living with HIV to know their status, second 90% of those who know their status are to be on treatment, and third 90% of those on treatment are to be virally suppressed [2].

A systematic review and meta-analysis conducted in 2017 in Ethiopia depicted that the pooled prevalence of immunological failure was estimated to be 15.3% by using WHO treatment failures criteria [10]. Children and adolescents are more likely to fail ART than adults. A retrospective cohort study conducted at Addis Ababa in four hospitals showed that 14.1% of children who were on ART had experienced a first-line treatment failure [11].

Even though, various global and local studies indicated a high proportion of treatment failure among children [11–13], little is known about the predictors of virological failure among children in Ethiopia. Most studies define treatment failure based on the two WHO criteria, clinical and immunological criteria. Currently, routine viral load workup for monitoring treatment failure is being implemented in Ethiopia. Therefore, the purpose of this study is, based on routine viral load measurement along with clinical and immunological criteria, to assess

the incidence and other underlying predictors of treatment failure among children on first-line antiretroviral therapy in Kolfe Keranyo sub-city Addis Ababa, Ethiopia.

## Materials and methods

### Study design, setting, and population

An institution-based retrospective cohort study was conducted from January to March 2021 among children enrolled in first-line ART from January 2013 to May 2020 in selected health facilities in Kolfe Keranyo sub-city. Kolfe Keranyo sub-city is one of the eleven sub-cities of Addis Ababa, the capital city of Ethiopia. Administratively, the sub-city is divided into 15 Woredas encompassing 1 hospital and 11 functional governmental health centers. The total population of Kolfe Keranyo sub-city in 2019 was estimated to be around 576,443. The sub-city is providing ART services to 14,161 people living with HIV. All children aged less than 15 years old who started receiving ART and are on follow-up at least for six months at the selected health facilities in Kolfe Keranyo sub-city from 2013 and 2020 were included in this study. Whereas, children with incomplete medical records were excluded [14].

### Sample size and sampling technique

The required sample size was determined by using a single population proportion formula based on a similar study conducted among the pediatric population in Ethiopia which showed a 17.3% prevalence of ART failure [15]. An assumption of a 5% margin of error, 95% confidence interval, and 80% of power were considered and the final sample size was 250.

The health facilities were selected based on their number of pediatric ART clients. Facilities with more than ten pediatric ART clients in a year were selected. From the 11 health centers and one hospital in the sub-city, five health centers and one hospital were selected. After preparing a sampling frame, the total sample size was then proportionally allocated to the selected health facilities. Finally, medical charts were selected with a simple random sampling technique (Fig 1).

### Data collection and quality control

A structured data abstraction checklist in English language containing the most relevant variables regarding ART treatment outcomes was adopted from different previous studies [1, 8, 11]. Data were collected both from patient follow-up charts and an electronic database. The data abstraction checklist included socio-demographic, ART, clinical, laboratory, and treatment-related characteristics. The data was retrieved by eight data clerks who have a diploma in information technology and experience in managing ART data in different facilities. One data collector was assigned to collect the data from each health center and three data collectors for the hospital. A one day of training on data collection tools and processes was conducted. The adopted data collection checklist was pretested to check for its completeness and consistency. The principal investigator and the assigned supervisors ensured the data completeness, accuracy, and consistency.

### Operational definitions

**Children**: Individuals with ages less than 15 years old.

**Virological failure**: Viral load above 1000 copies/mL based on two consecutive viral load measurements in 3 months, with adherence support following the first viral load test [2].

**Immunological failure**: Fall of CD4 count to baseline (or below) or 50% fall from on treatment peak value or persistent CD4 level below100 cells/mm3 [2].

**Clinical failure**: New or recurrent WHO clinical stage 3 and 4 condition.

**Treatment failure**:—Patients having clinical, immunological, or virological failures.

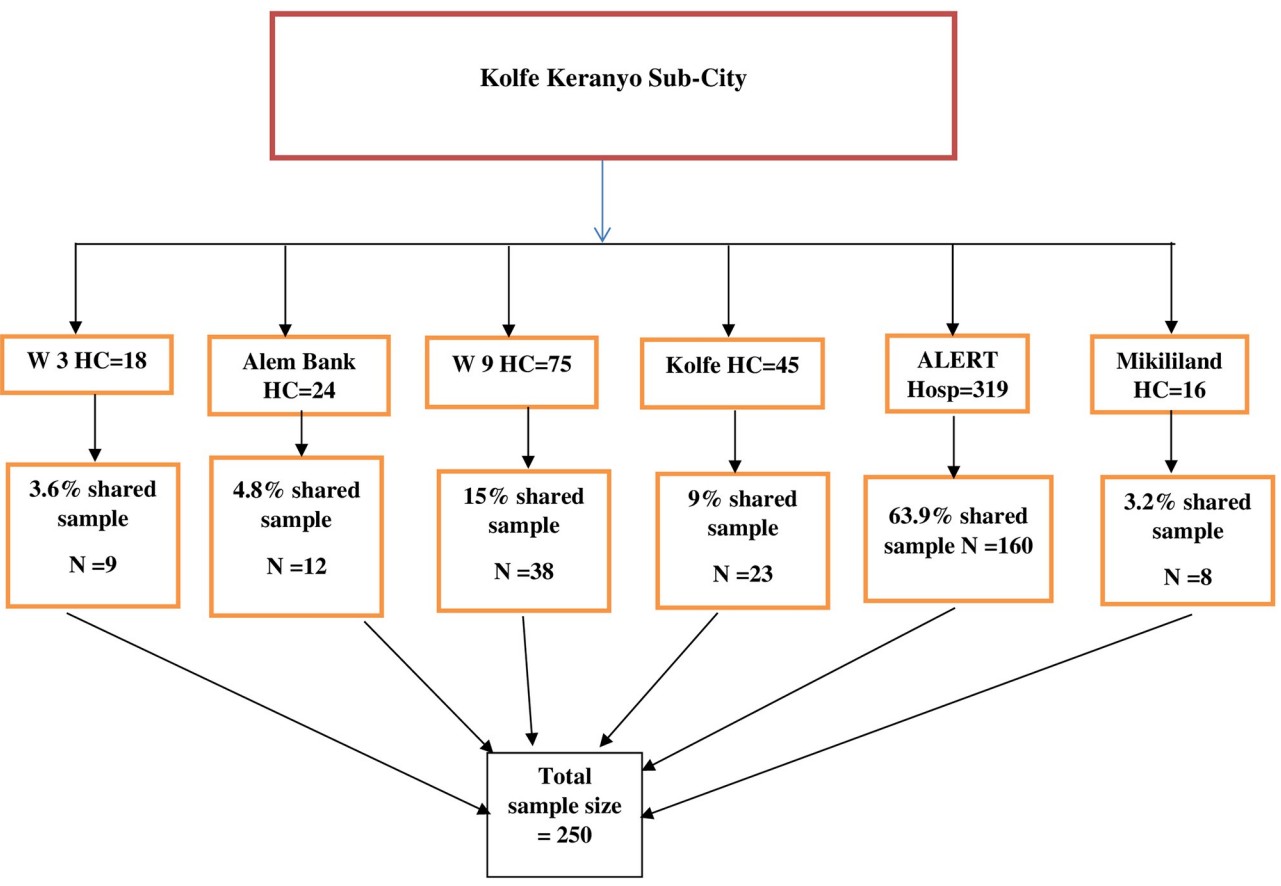

**Fig 1. Sample size distribution of study participants in Kolfe Keranyo sub-city, Addis Ababa, Ethiopia.**

**Adherence:** The extent to which the patient's behavior corresponds with the agreed upon recommendations from a health care provider [16].

**Adherence support:** A structured counseling by healthcare providers, peer counselors, using reminders, and other related activities to improve patient's adherence [16].

**Poor Adherence:** is defined in terms of total missed doses by greater than three days per month mainly based on pill count [2].

**Censored**: Patients who completed their follow-up, transferred out, or lost without developing virological failure.

**Time to detection of treatment failure**: The time between ART initiation and detection of treatment failure of first-line ART.

**Lost to follow-up:** Patients who have not received ART medication for more than 30 days but less than 90 days after their last missed drug collection appointment [2].

**Drop from follow-up:** Patients who have not received ART medication for more than 90 days of their last missed drug collection appointment [2].

**Drug substitution:** Replacement of at least one drug from the original ART regimen [2].

## Data processing and analysis

The collected data were entered into Epi Info version 7 and analyzed using SPSS version 26. Data cleaning was performed to check for frequencies, missed or error values, and identified

errors were corrected after revision of the original completed questionnaire. Descriptive statistics was used to describe the socio-demographic, clinical, laboratory, and treatment-related characteristics of patients. The Kaplan–Meier method was used to estimate the probability of treatment failure at different time points. Incidence of treatment failure was calculated per person-months of observation. To identify the predictors of the first-line treatment failure the Cox proportional hazards model was applied. First, a bivariable analysis was conducted and variables with p-value < 0.25 were entered into a multivariable analysis to control the effects of confounders. In the multivariable analysis, variables with p-value < 0.05 were declared as statistically significant with a 95% confidence interval.

## Ethical considerations

Ethical clearance was obtained from the Institution Review Board (IRB) of Addis Ababa Public Health Research and Emergency Management Directorate. Then, an official letter was submitted to the selected health facilities, and signed permission was obtained to access the ART client's database and charts. To maintain confidentiality patient names and unique ART numbers were not included in the data extraction checklist. Furthermore, the confidentiality of data was kept at all levels of the study and not used for any other purposes than the stated study objectives.

## Results

### Socio-demographic and clinical characteristics

A total of 250 individual medical charts were included in this study. Slightly more than half (55.2%) were female while the remaining 44.8% were male. This showed that almost an equivalent proportion of male and female sample was taken for the study. The mean age at the start of ART was 10.5 years (SD ± 3.36) with a minimum age of 1 and maximum age of 14 years. The majority of participants (58.8%) fell in the age group 10–14 years. The majority of the participants (80%) lived with at least one of their biological parents. About three-quarters (76.4%) of the caregivers had a positive HIV serologic status. The mean follow-up period after ART initiation was 49.2 months ranging from 6 to 95 months (Table 1).

### Baseline clinical and antiretroviral treatment characteristics

One hundred nineteen (47.6%) of study participants were initiated on AZT-3TC-NVP based ART regimen. The median baseline CD4 count was 461 cells/mm3 (IQR: 304–607). More than half (63.6%) of the study participants had an experience of regimen change during their ART follow-up time when the program shifted from D4T to AZT or TDF and NNRTI to DTG or LPV/r. Among total, 11.2% received PMTCT/ infant prophylaxis. Among the children on ART, half (50.4%) received Isoniazid prophylaxis during their follow-up time. The majority (88%) of the participants had access to viral load measurement at least once during their follow-up time (Table 2).

### Incidence of treatment failure

Among the 250 study subjects, 82.8% were right censored. It was found that a total of 43 subjects (17.2%) have had treatment failure. Of the failures, 62.8% were virological, 18.6% were immunological, and 2.3% were clinical failures. A mixed virological and immunological failure accounted for 16.3%. Among patients who had treatment failure, 69.8% had switched regimens to a second-line regimen (**Table 3**). The overall incidence rate was 3.45 (95% CI: 2.57–4.67) per 1000 person-months of observations. The incidence of virological failure was 2.74

**Table 1. Socio-demographic characteristics of children on ART at Kolfe Keranyo sub-city, Addis Ababa, Ethiopia from 2013 to 2020 (n = 250).**

| Variables | | Frequency | % |
|---|---|---|---|
| Gender | Male | 112 | 44.8% |
| | Female | 138 | 55.2% |
| Age (years) | < 5 | 27 | 11.8% |
| | 5–10 | 76 | 30.4% |
| | > 10 | 147 | 58.8% |
| Parent status | Both Alive | 133 | 53.2% |
| | Single Alive | 74 | 29.6% |
| | None of them | 43 | 17.2% |
| Parent taking ART | Yes | 171 | 68.4% |
| | No | 33 | 13.2% |
| | Unknown | 46 | 18.4% |
| Primary Caretaker | Mother | 99 | 39.6% |
| | Father | 15 | 6% |
| | Both parent | 86 | 34.4% |
| | Brother/Sister | 21 | 8.4% |
| | Other/relative | 29 | 11.6% |
| Caregiver HIV status | Positive | 191 | 76.4% |
| | Negative | 9 | 3.6% |
| | Unknown | 50 | 20.0% |
| Months on ART | < = 36 | 116 | 46.4% |
| | 37–60 | 106 | 42.4% |
| | > 60 | 28 | 11.2% |

(95% CI: 1.95–3.83) per 1000 person-months, immunological failure was 1.13 (95% CI: 0.67–1.91) per 1000 person-months and clinical failure was 0.08 (95% CI: 0.01–21.5) per 1000 person-months (Fig 2).

## Predictors of treatment failure

The bivariable Cox proportional hazard analysis showed that baseline WHO clinical stage III and IV, lack of drug substitution, parent's HIV status, infant prophylaxis at birth, AZT/3TC/NVP based initial drug regimen, parents taking ART, and episodes of poor adherence were the significant predictors of treatment failure.

In the multivariable Cox regression analysis, infant prophylaxis, drug substitutions, AZT/3TC/NVP based initial drug regimen, and poor adherence were the predictors of treatment failure. Accordingly, children who received infant prophylaxis at birth had about 3.6 times (Adjusted Hazard Ratio (AHR): 3.59, 95% CI: 1.65–7,82) higher risk of treatment failure when compared to those who haven't received infant prophylaxis. Children with drug substitutions were 82% less likely to suffer from treatment failure when compared to those without drug substitutions (AHR: 0.18, 95% CI: 0.09–0.37). Children who were on AZT/3TC/NVP based initial treatment regimen had about 2.3 times (AHR: 2.27, 95% CI: 1.14–4.25) higher risk of treatment failure when compared to those who were on other initial regimens. Moreover, children with more than 3 episodes of poor ART adherence had about 2.3 times (AHR: 2.27, 95% CI: 1.17–4.38) higher risk of treatment failure when compared to those with less than 3 episodes of poor adherence (Table 4).

**Table 2. Baseline clinical and antiretroviral treatment characteristics of children on first-line ART at Kolfe Keranyo sub-city, Addis Ababa, Ethiopia from 2013 to 2020 (n = 250).**

| Variables | | Frequency | % |
|---|---|---|---|
| WHO stage | Stage I | 141 | 56.4% |
| | Stage II | 65 | 26% |
| | Stage III | 36 | 14.4% |
| | Stage IV | 8 | 3.2% |
| Regimen at initiation | D4T/3TC/NVP | 14 | 5.6% |
| | D4T/3TC/EFV | 2 | 0.8% |
| | AZT/3TC/NVP | 119 | 47.6% |
| | AZT/3TC/EFV | 52 | 20.8% |
| | TDF/3TC/DTG | 14 | 5.6% |
| | ABC/3TC/LPV/r | 14 | 5.6% |
| | ABC/3TC/DTG | 7 | 2.8% |
| | AZT/3TC/LPV/r | 6 | 2.4% |
| | TDF/3TC/EFV | 20 | 8% |
| Infant Prophylaxis | Yes | 28 | 11.2% |
| | No | 222 | 88.8% |
| Disclosure of HIV status | Disclosed | 116 | 46.4% |
| | Not Disclosed | 106 | 42.4% |
| | Unknown | 28 | 11.2% |
| Status last visit | Alive | 220 | 88% |
| | Transfer Out | 15 | 6% |
| | Dead | 2 | 0.8% |
| | Drop[a] | 8 | 3.2% |
| | Lost[b] | 5 | 2% |
| Isoniazid Prophylaxis | Yes | 126 | 50.4% |
| | No | 124 | 49.6% |
| Regimen substitutions | Yes | 159 | 63.6% |
| | No | 91 | 36.4% |
| Viral load measurement | Yes | 220 | 88% |
| | No | 30 | 12% |

[a]**Drop:** Patients who have not received ART medication for more than 90 days of their last missed drug collection appointment.
[b]**Lost:** Patients who have not received ART medication for more than 30 days but less than 90 days after their last missed drug collection appointment.

**Table 3. Treatment failure rate and types of failures in children on first-line ART at Kolfe Keranyo sub-city, Addis Ababa, Ethiopia from 2013 to 2020 (n = 250).**

| Variables | | Frequency | % |
|---|---|---|---|
| First-line treatment failure | Yes | 43 | 17.2% |
| | No | 207 | 82.8% |
| Type of treatment failure | Clinical | 1 | 2.3% |
| | Virological | 27 | 62.8% |
| | Immunological | 8 | 18.6% |
| | Mixed (clinical and immunological) | 0 | 0% |
| | Mixed (virological and immunological) | 7 | 16.3% |
| | Mixed (three of them) | 0 | 0% |
| Switched to a second-line regimen | Change regimen | 30 | 69.8 % |
| | No change | 13 | 30.2% |

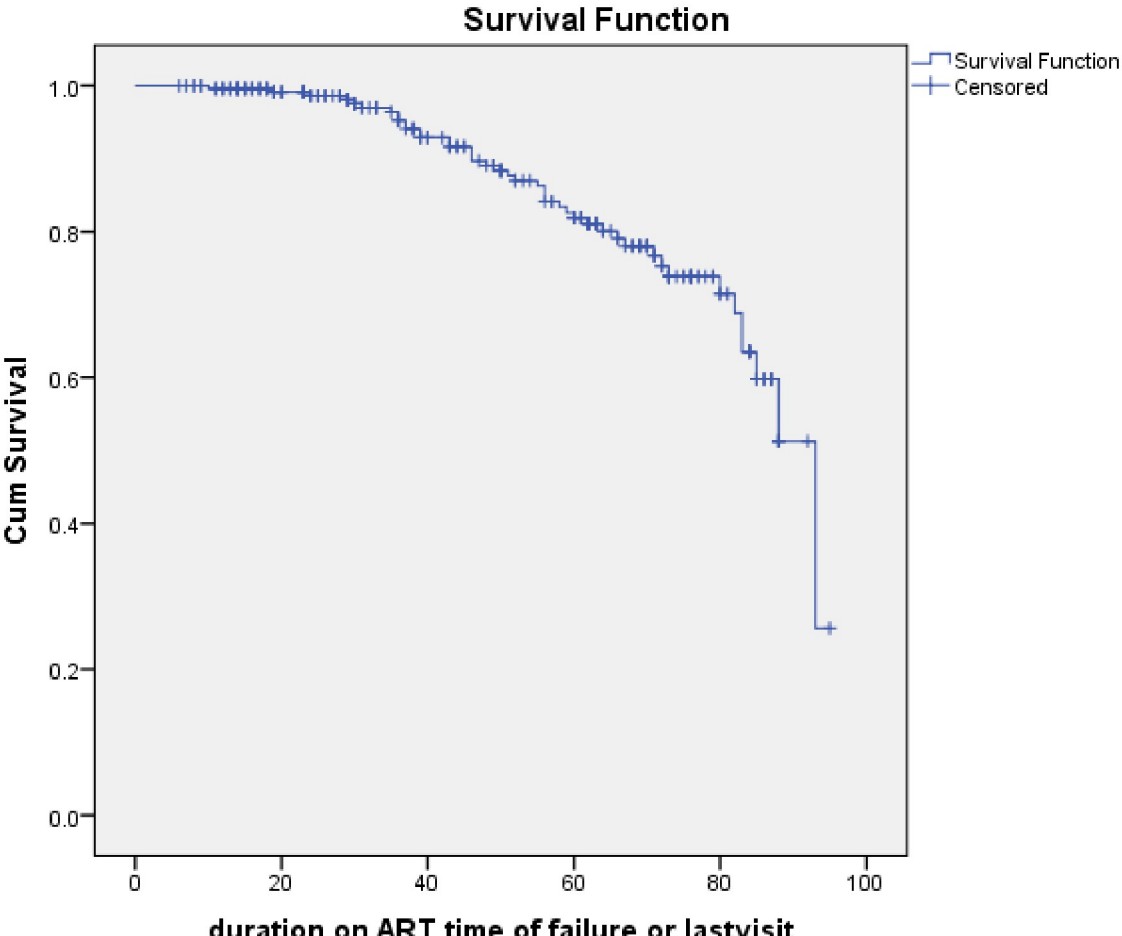

**Fig 2. The Kaplan Meier Survival function of children on first-line ART at Kolfe Keranyo sub-city, Addis Ababa, Ethiopia from 2013 to 2020 (n = 250).**

## Discussion

Despite the availability of ART drugs, antiretroviral treatment failure has been a concerning issue globally, especially in developing countries like Ethiopia [10, 17]. This study was conducted to assess the incidence and predictors of treatment failure among children on first-line antiretroviral therapy in Kolfe Keranyo sub-city, Addis Ababa, Ethiopia. In this study, a 17.2% treatment failure with 3.45 (95% CI: 2.57–4.67) per 1000 person-month incidence rate was observed. Moreover, infant prophylaxis, lack of drug substitutions, AZT/3TC/NVP based initial drug regimen, and poor adherence were the predictors of treatment failure.

The overall treatment failure in this study was 17.2%; virological, immunological, and clinical failure accounted for 79%, 18.6%, and 2.3% respectively. The current finding is comparable with the finding reported at Fiche and Kuyu hospital, 18.9% [18], Gondar hospital, 18.2% [19], and a study in Ethiopia, 17.3% [15]. On the other hand, treatment failure was found to be lower as compared to the findings reported in black lion hospital, 22.6% [20], Kenya, 37% [21], Uganda, 34% [9], and Zimbabwe, 30.6% [22]. However, it is higher than that of the study conducted in Amhara regional state, 7.7% [8], Amhara region referral hospitals, 12.1% [1], Felegehiwot referral hospital, 10.7% [23], India, 5% [24], and Addis Ababa, 14.1% [11].

**Table 4. Multivariable Cox regression analysis showing predictors of ART treatment failure at Kolfe Keranyo sub-city, Addis Ababa, Ethiopia from 2013 to 2020 (n = 250).**

| Variables | Treatment Failure | | AHR (95% CI) | p-value |
|---|---|---|---|---|
| | No | Yes | | |
| | n (%) | n (%) | | |
| **Infant prophylaxis** | | | | |
| No | 190 (85.6) | 32 (14.4) | 1 | |
| Yes | 17 (60.7) | 11 (39.3) | 3.59 (1.65–7.82) | **0.001*** |
| **Parent receiving ART** | | | | |
| No | 72 (91.1) | 7 (9.1) | 1 | |
| Yes | 135 (78.9) | 36 (21.1) | 0.87 (0.21–3.56) | 0.846 |
| **Drug substitution** | | | | |
| No | 63 (69.2) | 28 (30.8) | 1 | |
| Yes | 145 (91.2) | 14 (8.8) | 0.19 (0.09–0.37) | **<0.001*** |
| **Baseline WHO stage** | | | | |
| Stage I and II | 184 (85.2) | 32 (14.8) | 1 | |
| Stage III and IV | 24 (70.6) | 10 (29.4) | 0.81 (0.37–1.78) | 0.599 |
| **Initial ART regimen** | | | | |
| Other than AZT/3TC/NVP | 118 (90.1) | 13 (9.9) | 1 | |
| AZT/3TC/NVP | 89 (74.8) | 30 (24.2) | 2.27 (1.14–4.55) | **0.020*** |
| **Episodes of poor adherence** | | | | |
| <3 poor adherence | 206 (85.1) | 36 (14.9) | 1 | |
| >3 poor adherence | 1 (12.5) | 7 (87.5) | 2.27 (1.17–4.38) | **0.015*** |
| **Parent's HIV status** | | | | |
| Positive | 153 (80.1) | 38 (19.9) | 1 | |
| Negative | 54 (91.5) | 5 (8.5) | 1.77 (0.36–8.59) | 0.487 |

In this study, virological failure shared the highest number of failures followed by immunologic failure and only a small proportion of clinical failure which is similar to the findings in Amhara region referral hospital [1], Uganda [9], and Kenya [21]. The reason for the highest proportion of virological failure might be that virologic failure early detects treatment failure than immunological and clinical failures.

In this study PMTCT prophylaxis at birth was found as a predictor of treatment failure. Comparable findings were reported in studies conducted in Uganda [9] and Gondar [19]. NVP is the most commonly used drug during PMTCT and it is also used as a backbone for NNRTI regimen. More common adverse effects associated with NVP and its higher reported resistance [9] might contribute to a higher risk of treatment failure.

This study showed that there is a 5-fold risk of treatment failure at every point in the study if drug substitution does not occur. This finding contradicts with the study results in Addis Ababa [11], Gondar [19], and Felegehiwot hospital [23]. This might be due to the use of previous drugs for highly mutated viruses. Currently, most substitutions are provided by newly available drugs (protease inhibitors and integrase inhibitors) for optimized pediatric regimens.

AZT/3TC/NVP based initial regimen was also one of the predictors of treatment failure in this study. This is similar to the findings reported in studies conducted in Amhara region referral hospital [1] and Addis Ababa [11]. Similarly, studies conducted in Uganda [9] and Zimbabwe [22] reported NVP based regimen as a predictor of treatment failure. Reported lower efficacy of NVP when compared to EFV [9] could explain this.

Poor ART adherence among children on first-line ART was another predictor of treatment failure in this study. This finding is supported by various previous studies conducted in

Ethiopia [15], Uganda [9], Gondar [19], Felegehiwot hospital [23], Amhara region referral hospital [8], and Fiche and Kuyu hospital in Oromia region [18]. This could be explained by poor adherence highly contributed to viral mutations and resistance to antiretroviral drugs which could result in treatment failure.

Contrary to the findings in several other studies [11, 18, 23, 25], baseline CD4 count was not a predictor of treatment failure in this study. Whereas, consistent findings were reported in studies conducted in Amhara region in different hospitals [1, 8]. This variation could be explained by recently, ART is being provided for every HIV positive individual irrespective of their baseline CD4 count which could result in a better outcome and a lower chance to develop treatment failure.

Studies conducted in Amhara region referral hospitals [1], Black lion hospital [20], and Fiche and Kuyu [18] reported that WHO clinical stages and OIs as significant predictors of treatment failure. Whereas, this study reported a contradictory finding, which is consistent with the findings reported in Uganda [9]. This could be due to the studies conducted in Black lion hospital and Fiche and Kuyu were conducted before the test and treat strategy was implemented in Ethiopia which might result in higher number of children with advanced WHO clinical stages. The use of a relatively small sample size in this study could also be the reason for the observed differences.

Moreover, HIV status disclosure and the status of the primary caretaker were not predictors of treatment failure in this study. These findings were in contrast with the findings reported in Amhara region [8] and Black lion hospital [20]. These variations might be due to differences in sample size. Most of the children's caretakers were their own parents, this might also be another reason for the observed variations.

Although this is a retrospective cohort study, there were few incomplete documents that can be considered as a limitation. The temporal relationship between CD4 count determinations and concurrent illnesses was also difficult to determine due to the retrospective nature of the study. This study is not also free from selection bias, as a random selection was not used for sampling. Despite these limitations, the findings of this study could inform the national HIV programme for better practices and policies in pediatric HIV care.

## Conclusions

The proportion of treatment failure among children on first-line ART in Kolfe Keranyo sub-city in Addis Ababa was found to be high according to the UNAIDs virological suppression targets. Infant prophylaxis for PMTCT, lack of drug substitution, AZT/3TC/NVP based initial regimen, and poor adherence were found to be the predictors of first-line ART treatment failure. Hence, stakeholders working on HIV need to focus on these factors in order to minimize the rate of treatment failure.

Utilizing digital health tools to support ART adherence, such as reminder apps or text message reminders as well as creating support groups or peer networks for children and their caregivers to share their experiences and provide mutual support could be essential. National stakeholders might need to revise the AZT/3TC/NVP based regimen and compare it with other regimens for its effectiveness. Furthermore, future broader studies should examine these factors mainly the first-line regimen by using viral load as a measurement for treatment failure.

## Supporting information

**S1 File.**
(SAV)

## Acknowledgments

We would also like to express our gratitude to Addis Ababa City Administration Health Bureau and Kolfe Keranyo sub-city for allowing us to access the required patient data. We also would like to thank the data collectors and supervisors who contributed significantly to the success of this research work.

## Author Contributions

**Conceptualization:** Meseret Misasew.

**Data curation:** Meseret Misasew.

**Formal analysis:** Meseret Misasew, Takele Menna, Eyoel Berhan, Daniel Angassa, Yesunesh Teshome.

**Funding acquisition:** Meseret Misasew.

**Investigation:** Meseret Misasew, Takele Menna, Eyoel Berhan, Daniel Angassa, Yesunesh Teshome.

**Methodology:** Meseret Misasew, Takele Menna, Eyoel Berhan, Daniel Angassa, Yesunesh Teshome.

**Project administration:** Meseret Misasew.

**Resources:** Meseret Misasew.

**Software:** Meseret Misasew.

**Supervision:** Meseret Misasew.

**Validation:** Meseret Misasew, Takele Menna, Eyoel Berhan, Daniel Angassa, Yesunesh Teshome.

**Visualization:** Meseret Misasew.

**Writing – original draft:** Meseret Misasew, Takele Menna, Eyoel Berhan, Daniel Angassa, Yesunesh Teshome.

**Writing – review & editing:** Meseret Misasew, Takele Menna, Eyoel Berhan, Daniel Angassa, Yesunesh Teshome.

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
