## [Decision Letter · Decision Letter 0]

12 Jul 2022

PONE-D-22-08487Incidence and Predictors of Antiretroviral Treatment Failure among Children in Public Health Facilities of Kolfe Keranyo Sub-City, Addis Ababa, Ethiopia: Institution-based retrospective cohort studyPLOS ONE

Dear Dr. Misasew,

Thank you for submitting your manuscript to PLOS ONE. After careful consideration, we feel that it has merit but does not fully meet PLOS ONE’s publication criteria as it currently stands. Therefore, we invite you to submit a revised version of the manuscript that addresses the points raised during the review process.

Please consider comments from both reviewers. 

We look forward to receiving your revised manuscript.

Kind regards,

Chinmay Laxmeshwar

Academic Editor

PLOS ONE

Journal Requirements:

 “The authors have not received any finical support from any institution/government; hence, the operation of the research is solely financed by the handling fee received from the authors.”

Additional Editor Comments:

1. Please describe the standard treatment guidelines in the country and whether the regimen prescribed for patients in your study differed.

2. Isoniazid prophylaxis – is it initiation of completion of IPT? Also, how long ago was this IPT prescribed?

3. Were any patients’ regimen was switched due to failure?

4. Please proofread the manuscript and check the typos and grammar.

Reviewers' comments:

Reviewer's Responses to Questions

**Comments to the Author**

1. Is the manuscript technically sound, and do the data support the conclusions?

Reviewer #1: Yes

Reviewer #2: Partly

2. Has the statistical analysis been performed appropriately and rigorously? 

Reviewer #1: I Don't Know

Reviewer #2: Yes

3. Have the authors made all data underlying the findings in their manuscript fully available?

Reviewer #1: Yes

Reviewer #2: Yes

4. Is the manuscript presented in an intelligible fashion and written in standard English?

Reviewer #1: Yes

Reviewer #2: Yes

5. Review Comments to the Author

Reviewer #1: I wrote some minor comments in attached document, so my main suggestions are highlight in yellow with a comment attached. I think the manuscrit is technicaly sound and data presented supports the findings. I do not see any possible ethical conflict in it. I am afraid I cannot comment on the statistical analyses as I do not have enough knowledge about the same.

Reviewer #2: Thank you for the very interesting manuscript. I would like to suggest a some revisions to make it clearer to readers.

1. Introduction:

It would be interesting to include a couple of sentences on why you decided to carry out this study and what you hope to achieve based on the conclusions drawn.

2. Methods:

i) As regards ethics, please mention specifically why consent of patients was waived for this study.

ii) Why is your sample size derived proportionally as opposed to including all participants that meet selection criteria?

iii) Please reference the studies from which you derived your questionnaire.

iv) How was your data anonymized and who had access to the key? Also, can you explain how you dealt with other identifying factors such as date of birth and address?

v) Please indicate how bias was addressed. I see some sources. For example: You select centres with >10 children. This means that they are slightly larger centres, probably with more expertise. Also, I believe that the test and treat strategy was implemented in Ethiopia only after 2017, so this might bias results.

vi) The definition of adherence is unclear. Why are missed appointments mentioned? How is adherence measured in the study.

vii) Please define drug substitution in the operational definitions. Also, it would be good to have a reference of which drugs are used for EMTCT.

viii) I see that you have defined the p value as 0.25 for bivariate analysis. Can you explain why you have chosen this threshold?

3. Results:

i) Can you explain how you dealt with missing data?

ii) I suggest you stratify by patients before and after the test and treat strategy was implemented in Ethiopia. This would be an effect modifier for the baseline CD4

iii) I see you mention virological failure in the text of your results but I do not see it in the table. Could you include that variable to, especially since you specifically refer to it in the introduction.

Discussion:

1. You mention poor ART adherence as a predictor of treatment failure. However, your definition mentions appointments while the other studies mentioned discuss treatment adherence. Please clarify.

2. Please mention any biases in your limitations.

3. It is important to mention the generalisability of your findings. How does it add to the body of knowledge on the topic at a regional and international level

Conclusion:

I would suggest that you focus on factors deemed significant by your analysis in the conclusion, otherwise you run the risk of it sounding like a generic wrap-up. Focus on important factors will lead to more of a connect between the results and the conclusion.

A last comment: I saw some grammatical and syntactical errors in the article.

In summary, I believe this article has scientific relevance and look forward to reading it when it is published.

6. PLOS authors have the option to publish the peer review history of their article (what does this mean?). If published, this will include your full peer review and any attached files.

Reviewer #1: No

Reviewer #2: **Yes: **Parvati Nair

---

## [Author Response · Author response to Decision Letter 0]

15 Aug 2022

Response to Reviewers

Title: Incidence and Predictors of Antiretroviral Treatment Failure among Children in Public Health Facilities of Kolfe Keranyo Sub-City, Addis Ababa, Ethiopia: Institution-based retrospective cohort study

Authors:

Meseret Misasew (mesimisasew@gmail.com)

Daniel Angassa (danielssa25@gmail.com)

Yesunesh Teshome (yesunesh.teshome@yahoo.com)

Version: 1 Date: 15 August 2022 

Response to reviewers’ comments and suggestions

We would like to thank both the editor and reviewers for taking the time to read our paper and forward their valuable comments and suggestions which in turn helped us to make a significant improvement to our manuscript. We have attempted to address all the reviewers' comments and suggestions and carefully took corrective measures. Additionally, we have changed the affiliation of one author, Daniel Angassa, and included one additional author, Yesunesh Teshome, full information including affiliation is included in the revised manuscript. These changes can be found highlighted in light blue color in the title page and Authors’ contribution section of the revised manuscript, lines 5, 8, 10, 291, 293, 294, 299, 301, and 302.

Furthermore, the point-by-point response to the editor’s and reviewers’ comments is presented below. Our responses are written in blue font color. 

Academic Editor:

i. Please describe the standard treatment guidelines in the country and whether the regimen prescribed for patients in your study differed.

Authors’ response: The standard treatment guideline implemented in Ethiopia states that all HIV positives are eligible for ART. The preferred first-line regimen for children and adolescents is AZT or ABC + 3TC + EFV. The alternative first-line regimes are AZT + 3TC + NVP, TDF + 3TC + EFV, TDF + 3TC + NVP, and ABC + 3TC + NVP. The aforementioned regimes are being implemented all over the country and the regimens prescribed in our study were similar to those regimens. 

ii. Isoniazid prophylaxis – is it initiation of completion of IPT? Also, how long ago was this IPT prescribed? 

Authors’ response: It is the initiation of IPT and IPT was prescribed for at least six months, as the national treatment guideline recommends.

iii. Were any patients’ regimen was switched due to failure?

Authors’ response: Yes, among patients who had treatment failure, 69.8% had switched regimen to a second-line regimen. We have included this sentence and presented it in a table in the results section of the revised manuscript. These changes can be found highlighted in the results section of the revised manuscript, lines 181-182.

iv. Please proofread the manuscript and check the typos and grammar.

Authors’ response: Thank you for your valuable comment. As suggested, we proofread and made significant grammatical and technical corrections to all parts of the manuscript.

Reviewer # 1:

1. Abstract: 

i. The sentence is not very clear, this is 53% from the ones who are on ART? or from the total number of HIV infected? 

ii. Which initial regimen?

Authors’ response: Thank you for raising these questions. 

i. On the selected sentence, 53% represents those who are on ART and had achieved viral suppression. We have improved the sentence in order to make it clear to readers. 

ii. The initial regimen mentioned in this section is AZT/3TC/NVP based regimen and we have included it to make it more clear to readers.

These changes can be found highlighted in the abstract section of the revised manuscript, lines 19 and 38.

2. Introduction: 

i. It is also very low for adults, in this sentence seams that 61% coverage is acceptable but is not as is very far from the target of 90%. 

ii. Will be more interesting to know the % of children failing treatment.

iii. Since when is implemented? as this study review data from 2013 to 2020 the assessment of treatment failure will be diferent before and after implementation of routine viral load. 

Authors’ response: Thank you again for your valuable comments. 

i. As you stated 61% ART coverage for adults is also low as compared to the target, 90%. Hence, we have improved the sentence in a way that makes more sense. 

ii. In the selected sentence, 14.1% was intended to represent the percentage of children with treatment failures and we have improved the sentence in a better meaningful way.

iii. Routine viral load workup for monitoring treatment failure is being implemented in Ethiopia, since 2016. We have used viral load measurement along with clinical and immunological criteria to assess treatment failure in our study.

These changes can be found highlighted in the introduction section of the revised manuscript, lines 51-53 and 67-68.

3. Materials and Methods:

i. To mention somewhere definition of adherence support, as structured talk with a counselor, use of reminders, or any other strategy.....

ii. This is appointment to health center? how many appointments had the patients per month??.....miss dosages are not counted as poor adherence?

Authors’ response: Thank you again for the important suggestions. 

i. We have included definitions of the terms ‘Adherence’ and ‘Adherence Support’. 

ii. Since counting missed doses is the correct way of monitoring adherence, we have also changed the word ‘appointments’ into ‘doses’ on the definition of poor adherence.

These changes can be found highlighted in the materials and methods section of the revised manuscript, lines 120-123.

4. Results:

i. Why to say during follow up time, is NVP prophylaxis is done at birth before decision of starting ART?

ii. What is the difference between drop and lost?

iii. Which one?

Authors’ response: Thank you again for raising these important questions. 

i. NVP prophylaxis is given to all infants born to HIV infected mothers within one hour of birth before the decision of starting ART. 

ii. We have included the definitions of ‘Lost’ and ‘Drop’ in the operational definitions. 

iii. The mentioned initial regimen is AZT/3TC/NVP based regimen and we have included it in every section of the revised manuscript. 

These changes can be found highlighted in the materials and methods, and results section of the revised manuscript, lines 130-133, 191,195, and 213.

5. Discussion

i. According to description above is the lack of drug substitution what is a predictor of treatment failure.... the way is written is a bit confusing to me.

ii. The 250 children had access to viral load? , if not perhaps somewhere to describe how many were assessed with viral load.

iii. Was viral load assessed before drug substitution? otherwise the initial treatment before the substitution could be failed without knowing it.

Authors’ response: Thank you again for your valuable comments. 

i. Lack of drug substitution was the predictor of treatment failure and we have improved the sentence accordingly on every section of the revised manuscript. 

ii. The number of children who had access to viral load measurement was 220 (88%) and we have included this sentence on the revised manuscript.

iii. Initial viral load measurement was done before drug substitution which enabled us to differentiate which drug regimen failed. 

These changes can be found highlighted in the results and discussion sections of the revised manuscript, lines 173-174, 190-191, and 213.

6. Conclusions:

i. Same comments than before.

ii. What factors?

Authors’ response: Thank you again for your valuable comments. 

i. We accepted the comment and improved the sentence in the revised manuscript. 

ii. As explained in the introduction section of the manuscript, literature regarding predictors of virological failure among children in Ethiopia is scarce. Therefore, we are suggesting future broader studies to assess other relevant factors including viral load as a measurement for treatment failure. We have amended accordingly some sentences in the conclusion section of the revised manuscript. 

These changes can be found highlighted in the conclusions section of the revised manuscript, lines 274, 281, and 282.

Reviewer # 2:

i. Introduction:

It would be interesting to include a couple of sentences on why you decided to carry out this study and what you hope to achieve based on the conclusions drawn.

Authors’ response: Thank you for your valuable comment. We accepted the comment and amended the suggested sentence in the introduction section of the revised manuscript. This change can be found highlighted in the introduction section of the revised manuscript, lines 73-76.

ii. Methods:

i. As regards ethics, please mention specifically why consent of patients was waived for this study.

ii. Why is your sample size derived proportionally as opposed to including all participants that meet selection criteria?

iii. Please reference the studies from which you derived your questionnaire.

iv. How was your data anonymized and who had access to the key? Also, can you explain how you dealt with other identifying factors such as date of birth and address?

v. Please indicate how bias was addressed. I see some sources. For example: You select centres with >10 children. This means that they are slightly larger centres, probably with more expertise. Also, I believe that the test and treat strategy was implemented in Ethiopia only after 2017, so this might bias results.

vi. The definition of adherence is unclear. Why are missed appointments mentioned? How is adherence measured in the study?

vii. Please define drug substitution in the operational definitions. Also, it would be good to have a reference of which drugs are used for EMTCT.

viii. I see that you have defined the p value as 0.25 for bivariate analysis. Can you explain why you have chosen this threshold?

Authors’ response: Thank you again for your valuable comments and questions. 

i. Since we have used secondary data and there was no contact with patients, the selected health facilities have signed consent and gave permission to access the ART patient’s database and charts rather than obtaining consent of patients.

ii. We have derived our sample size proportionally rather than including all participants who meet the selection criteria due to budget and resource constraints since the authors haven’t received any financial support or fund.

iii. We have cited the studies from which we adopted our questionnaire and put them in the reference.

iv. Our data was anonymized by not including patient names and unique ART numbers in the data extraction checklist and only the authors had access to the original dataset. We have included date of birth/ age but not address of patients in the data set.

v. Thank you and we have included selection bias in our limitations. We have tried to minimize selection bias by increasing the number of health facilities that were included in the sampling frame, about half of the health centers were included. The only hospital with a large number of patients was also included.

vi. As explained above we have amended the definition of adherence and rather than missed appointments we have used missed doses to measure adherence. 

vii. We have included drug substitution in the operational definition as suggested. The drugs used for EMTCT are NVP and AZT.

viii. Since we sustain a shortfall of predictor variables at p-value 0.05 in the bivariate analysis, we have chosen a p-value of 0.25, a commonly used cut-off value, for variables to be included in the multivariable analysis. 

These changes can be found highlighted in the materials and methods section of the revised manuscript, lines 102, 120, 121, 124, 134, 268-270.

iii. Results:

i. Can you explain how you dealt with missing data?

ii. I suggest you stratify by patients before and after the test and treat strategy was implemented in Ethiopia. This would be an effect modifier for the baseline CD4

iii. I see you mention virological failure in the text of your results but I do not see it in the table. Could you include that variable to, especially since you specifically refer to it in the introduction?

Authors’ response: Thank you again for your questions and important suggestions.

i. We have encountered only a few missed data and hence, we omitted those few missed data and analyzed the remaining data.

ii. We were not able to stratify patients as such because baseline CD4 assessment was not conducted for all patients before and after the test and treat strategy was implemented. There were times when CD4 machines were not fully functional all over the country.

iii. We have included a table that presented types of treatment failure including virological failure, as suggested. 

These changes can be found highlighted in the materials and methods, and results sections of the revised manuscript, lines 187-188.

iv. Discussion:

iv. You mention poor ART adherence as a predictor of treatment failure. However, your definition mentions appointments while the other studies mentioned discuss treatment adherence. Please clarify.

v. Please mention any biases in your limitations.

vi. It is important to mention the generalisability of your findings. How does it add to the body of knowledge on the topic at a regional and international level?

Authors’ response: Thank you again for your important suggestions and questions.

i. We have amended the definition of adherence in the revised manuscript. 

ii. We have included selection bias in our limitation. 

iii. As suggested, we have included a sentence about the generalizability of our findings.

These changes can be found highlighted in the materials and methods and discussion sections of the revised manuscript, lines 120-121, and 268-270.

v. Conclusion:

i. I would suggest that you focus on factors deemed significant by your analysis in the conclusion, otherwise you run the risk of it sounding like a generic wrap-up. Focus on important factors will lead to more of a connect between the results and the conclusion.

Authors’ response: Thank you again for your suggestion. We tried to amend the conclusion in line with the significant factors from our results, as suggested. These changes can be found highlighted in the conclusions section of the revised manuscript, lines 278-281.

vi. A last comment: I saw some grammatical and syntactical errors in the article.

Authors’ response: Thank you again and we have made significant grammatical and syntactical corrections to all parts of the manuscript. 

vii. In summary, I believe this article has scientific relevance and look forward to reading it when it is published. 

Authors’ response: Thank you for your kind words and we have amended every section of the manuscript based on the reviewers’ and editor’s comments and suggestions.

Kind regards,

Meseret Misasew

Corresponding author

---

## [Decision Letter · Decision Letter 1]

7 Oct 2022

PONE-D-22-08487R1Incidence and Predictors of Antiretroviral Treatment Failure among Children in Public Health Facilities of Kolfe Keranyo Sub-City, Addis Ababa, Ethiopia: Institution-based retrospective cohort studyPLOS ONE

Dear Dr. Misasew,

Thank you for submitting your manuscript to PLOS ONE. After careful consideration, we feel that it has merit but does not fully meet PLOS ONE’s publication criteria as it currently stands. Therefore, we invite you to submit a revised version of the manuscript that addresses the points raised during the review process.

We look forward to receiving your revised manuscript.

Kind regards,

Chinmay Laxmeshwar

Academic Editor

PLOS ONE

Reviewers' comments:

Reviewer's Responses to Questions

**Comments to the Author**

1. If the authors have adequately addressed your comments raised in a previous round of review and you feel that this manuscript is now acceptable for publication, you may indicate that here to bypass the “Comments to the Author” section, enter your conflict of interest statement in the “Confidential to Editor” section, and submit your "Accept" recommendation.

Reviewer #2: (No Response)

Reviewer #3: (No Response)

2. Is the manuscript technically sound, and do the data support the conclusions?

Reviewer #2: Partly

Reviewer #3: Partly

3. Has the statistical analysis been performed appropriately and rigorously? 

Reviewer #2: Yes

Reviewer #3: Yes

4. Have the authors made all data underlying the findings in their manuscript fully available?

Reviewer #2: Yes

Reviewer #3: Yes

5. Is the manuscript presented in an intelligible fashion and written in standard English?

Reviewer #2: Yes

Reviewer #3: No

6. Review Comments to the Author

Reviewer #2: Thank you for addressing most of my many questions.

1. The manuscript is presented in an intelligible fashion, but there are many grammatical errors.

2. Results from the table are repeated in the manuscript text; value of repetition debatable

3. I still have a few concerns around phrasing:

i) Line 156 says "A total of 250 medical charts were included in this study with a 100% response rate". To me. response rate=survey

ii) Line 78 says: "82.8% were right censored (free of treatment failure)"  This is not the standard definition of right censoring and censoring is defined differently earlier.

iii) Does AHR mean adjusted Hazard Ratio? It is not expanded at first occurrence, hence unclear.

iv) Line 234: This might be due to the previous use of highly mutated drugs as substitute drugs.  Must rephrase; virus is mutated, not the drugs.

4. Line 232: "This study showed that drug substitution was about 18% protective for treatment failure"  Please check this, I think it is more that there is a 5-fold risk of failure at every point in the study if drug substitution does not occur.

Reviewer #3: 1. The clarification about ART coverage in line 51-53 still confuse. Need to be specified the source of 90% target, UNAIDS target. E.g. According to the 2017 Ministry of Health estimate, 722,248 Ethiopians are currently living with HIV but with a low ART coverage of 61% for adults and 33% for children, far from the UNAIDS target of 90%

2. Treatment failure in lines 59 is classified in so called later in the table as types, but in the text said is "measured" which doesn't seems accurate. And later refers to WHO criteria (line 71). May be harmonize IN one definition, clasiification and/or criterias

3. "This showed that a good proportion of male and female sample was taken for the study", Do you mean that both genders are equally represented? "Good proportion" is not an adequated vocabullary

4. "One hundred nineteen (47.6%) of study participants were found on AZT-3TC-NVP ART regimen during ART initiation":...were initiated on xxxx regimen"

5. Do you have the CD4 %? Being many U5 children, these seems important to be reported

6. "During their follow–up time, the majority of the study participants (88.8%) did not take PMTCT prophylaxis, even if born to HIV infected mothers". This refers to the patient follow-up? And applies to ones who were recruited immediately after born becasue if not how to value the PMTCT prophylxis? I'd suggest to say: "among total....xxx% received PMTCT/infant prophylaxis"

7. 189-Predictors of treatment failure: there is a confusion with "regimen substitution" (line 191) with "drug substitution" (line 194. And when is talkjing about "drug substitution" is the shifts that Programme did d4T to AZT, EFV to DTG, etc? It will be good also to clarify if these shifts were done using VL previous shifting or not

8. For Poor Adherence definition, please clarify is the missed doses are self-reported, by pill count or which other method

In Discussion

9. "The reason for the highest proportion of virological failure might be that virologic failure early detects treatment failure than immunological and clinical failure." And in many places is related to the lack of access to VL

10. "NVP is the most commonly used drug during PMTCT and it is also used as a backbone for NNRTI regimen (of HAART-I'd take out this) which might result in a higher risk of treatment failure." Please elaborate more this idea

11. "This study showed that drug substitution was about 18% protective for treatment failure" I don't understand this statement

12. "The fact that AZT/3TC/NVP regimen is not being used currently could explain the observed variations". Which variations

13. "This variation could be explained by recently, ART is being provided for every HIV positive individual and baseline CD4 is not being done routinely for each patient". This can explain lack of data but not lack of association

14. Line 252-259: these paragraph need to be rephrase. And not clear how T&T is linked with no having relation between low CD4 and failure

15. "Despite these limitations, the findings of this study could 269 be a great input for the national ART treatment program" I'd suggest : Could inform natioanal HIV Programme for better practices and policies in pediatric HIV care

16. Line 273: UNAIDS virological supression targets

Conclusion

17. "Health care providers should provide a timely drug substitution when required". This is applicable for treatment/regimen change when failure is detected which is different from the "drug substitution" you defined as punctual replacement of one drug becasue programmatic issues

18. In general the conclusions need to be described in a more objective way, not as an advise for parents/stakeholders/programme

7. PLOS authors have the option to publish the peer review history of their article (what does this mean?). If published, this will include your full peer review and any attached files.

Reviewer #2: **Yes: **Parvati Nair

Reviewer #3: No

---

## [Author Response · Author response to Decision Letter 1]

3 Nov 2022

Response to Reviewers

Title: Incidence and Predictors of Antiretroviral Treatment Failure among Children in Public Health Facilities of Kolfe Keranyo Sub-City, Addis Ababa, Ethiopia: Institution-based retrospective cohort study

Authors:

Meseret Misasew (mesimisasew@gmail.com)

Daniel Angassa (danielssa25@gmail.com)

Yesunesh Teshome (yesunesh.teshome@yahoo.com)

Version: 2 Date: 3 November 2022

Response to reviewers’ comments and suggestions

We would like to thank the reviewers for taking the time to read our paper and forward their valuable comments and suggestions which in turn helped us to make a significant improvement to our manuscript. We have attempted to address all the reviewers' comments and suggestions and carefully took corrective measures.

Furthermore, the point-by-point response to the reviewers’ comments is presented below. Our responses are written in blue font color. 

Reviewer # 2:

1. The manuscript is presented in an intelligible fashion, but there are many grammatical errors.

Authors’ response: Thank you for your valuable comment. We have made significant grammatical corrections to all parts of the manuscript.

2. Results from the table are repeated in the manuscript text; value of repetition debatable. 

Authors’ response: Thank you again for your comment. In the manuscript text, we have tried to narrate the main /predominant results presented in the tables.

3. I still have a few concerns around phrasing:

i. Line 156 says "A total of 250 medical charts were included in this study with a 100% response rate". To me. response rate=survey

ii. Line 178 says: "82.8% were right censored (free of treatment failure)"  This is not the standard definition of right censoring and censoring is defined differently earlier.

iii. Does AHR mean adjusted Hazard Ratio? It is not expanded at first occurrence, hence unclear.

iv. Line 234: This might be due to the previous use of highly mutated drugs as substitute drugs.  Must rephrase; virus is mutated, not the drugs.

Authors’ response: Thank you again for the important comments and suggestions. 

i. We have removed the phrase ‘with a 100% response rate’ from the mentioned sentence.

ii. We have removed the phrase ‘free of treatment failure’ from the mentioned sentence to avoid inconsistency with its operational definition.

iii. We have expanded AHR at the first occurrence to make it clear. 

iv. We have rephrased the mentioned sentence as suggested.

These changes can be found highlighted in the results section of the revised manuscript, lines 157, 196, 235, and 236.

4. Line 232: "This study showed that drug substitution was about 18% protective for treatment failure"  Please check this, I think it is more that there is a 5-fold risk of failure at every point in the study if drug substitution does not occur.

Authors’ response: Thank you again for raising this important suggestion. We have rephrased the mentioned sentence as suggested. This change can be found highlighted in the results section of the revised manuscript, lines 233-234.

Reviewer # 3:

1. The clarification about ART coverage in line 51-53 still confuse. Need to be specified the source of 90% target, UNAIDS target. E.g. According to the 2017 Ministry of Health estimate, 722,248 Ethiopians are currently living with HIV but with a low ART coverage of 61% for adults and 33% for children, far from the UNAIDS target of 90%. 

Authors’ response: Thank you for your valuable comment. We accepted the comment and amended the suggested sentence in the introduction section of the revised manuscript. This change can be found highlighted in the introduction section of the revised manuscript, lines 51-53.

2. Treatment failure in lines 59 is classified in so called later in the table as types, but in the text said is "measured" which doesn't seems accurate. And later refers to WHO criteria (line 71). May be harmonize IN one definition, classification and/or criteria.

Authors’ response: Thank you again for your valuable comment. We amended the mentioned sentence to make it consistent with the definitions we have used in the operational definition. We have used the standard WHO definitions for all types of treatment failures as mentioned in our operational definition. This change can be found highlighted in the introduction section of the revised manuscript, line 59.

3. "This showed that a good proportion of male and female sample was taken for the study", Do you mean that both genders are equally represented? "Good proportion" is not an adequate vocabulary.

Authors’ response: Thank you again for your valuable comment. We amended the mentioned sentence as suggested. This change can be found highlighted in the results section of the revised manuscript, line 157.

4. "One hundred nineteen (47.6%) of study participants were found on AZT-3TC-NVP ART regimen during ART initiation":...were initiated on xxxx regimen"

Authors’ response: Thank you again for your valuable comment. We amended the mentioned sentence as suggested. This change can be found highlighted in the results section of the revised manuscript, line 167.

5. Do you have the CD4 %? Being many U5 children, these seems important to be reported.

Authors’ response: Thank you again for your important suggestion and we completely agree that it is important to report DC4 % for the under 5 children. However, we were not able to report the CD4 % for under 5 children since the CD4 count was not available for each patient.

6. "During their follow–up time, the majority of the study participants (88.8%) did not take PMTCT prophylaxis, even if born to HIV infected mothers". This refers to the patient follow-up? And applies to ones who were recruited immediately after born becasue if not how to value the PMTCT prophlxis? I'd suggest to say: "among total....xxx% received PMTCT/infant prophylaxis"

Authors’ response: Thank you again for your important suggestion. We rephrased the mentioned sentence as suggested. This change can be found highlighted in the results section of the revised manuscript, lines 170-171.

7. 189-Predictors of treatment failure: there is a confusion with "regimen substitution" (line 191) with "drug substitution" (line 194. And when is talking about "drug substitution" is the shifts that Programme did d4T to AZT, EFV to DTG, etc? It will be good also to clarify if these shifts were done using VL previous shifting or not.

Authors’ response: Thank you again for your valuable comment. We amended the mentioned sentence as suggested. This change can be found highlighted in the results section of the revised manuscript, line 190.

8. For Poor Adherence definition, please clarify is the missed doses are self-reported, by pill count or which other method.

Authors’ response: Thank you again for your important suggestion. We amended the mentioned sentence as suggested. This change can be found highlighted in the materials and methods section of the revised manuscript, line 125.

9. "The reason for the highest proportion of virological failure might be that virologic failure early detects treatment failure than immunological and clinical failure." And in many places is related to the lack of access to VL.

Authors’ response: Thank you again for your comment. Lack of access to viral load was a problem previously. As mentioned in our introduction, lines 71-72, currently, routine viral load workup for monitoring treatment failure is being implemented in Ethiopia.

10. "NVP is the most commonly used drug during PMTCT and it is also used as a backbone for NNRTI regimen (of HAART-I'd take out this) which might result in a higher risk of treatment failure." Please elaborate more this idea.

Authors’ response: Thank you again for your important suggestion. We amended the mentioned sentence in a more descriptive form. This change can be found highlighted in the discussion section of the revised manuscript, lines 231-232.

11. "This study showed that drug substitution was about 18% protective for treatment failure" I don't understand this statement.

Authors’ response: Thank you again for your important comment. We have rephrased the mentioned sentence as suggested. This change can be found highlighted in the discussion section of the revised manuscript, lines 233-234.

12. "The fact that AZT/3TC/NVP regimen is not being used currently could explain the observed variations". Which variations

Authors’ response: Thank you again for your valuable comment. We have corrected the mentioned sentence. This change can be found highlighted in the discussion section of the revised manuscript, lines 241-242.

13. "This variation could be explained by recently, ART is being provided for every HIV positive individual and baseline CD4 is not being done routinely for each patient". This can explain lack of data but not lack of association

Authors’ response: Thank you again for your valuable comment. We have rephrased the mentioned sentence in a more meaningful way. This change can be found highlighted in the discussion section of the revised manuscript, lines 252-253.

14. Line 252-259: these paragraph need to be rephrase. And not clear how T&T is linked with no having relation between low CD4 and failure

Authors’ response: Thank you again for your valuable comment. We have rephrased the mentioned paragraph to make it clearer. This change can be found highlighted in the discussion section of the revised manuscript, lines 257-259.

15. "Despite these limitations, the findings of this study could 269 be a great input for the national ART treatment program" I'd suggest : Could inform natioanal HIV Programme for better practices and policies in pediatric HIV care.

Authors’ response: Thank you again for your important suggestion. We have rephrased the mentioned sentence as suggested. This change can be found highlighted in the discussion section of the revised manuscript, lines 270-271.

16. Line 273: UNAIDS virological supression targets

Authors’ response: Thank you again for your important suggestion. We have rephrased the mentioned sentence as suggested. This change can be found highlighted in the conclusions section of the revised manuscript, line 274.

17. "Health care providers should provide a timely drug substitution when required". This is applicable for treatment/regimen change when failure is detected which is different from the "drug substitution" you defined as punctual replacement of one drug becasue programmatic issues

Authors’ response: Thank you again for your valuable comment. We have omitted the sentence about drug substitution since it is associated with programmatic issues.

18. In general the conclusions need to be described in a more objective way, not as an advise for parents/stakeholders/programme

Authors’ response: Thank you again for your significant comment. We have rephrased the conclusion in a more objective way as suggested. These changes can be found highlighted in the conclusions section of the revised manuscript, lines 279-282.

Kind regards,

Meseret Misasew

Corresponding author

---

## [Decision Letter · Decision Letter 2]

27 Feb 2023

PONE-D-22-08487R2Incidence and Predictors of Antiretroviral Treatment Failure among Children in Public Health Facilities of Kolfe Keranyo Sub-City, Addis Ababa, Ethiopia: Institution-based retrospective cohort studyPLOS ONE

Dear Dr. Misasew,

Thank you for submitting your manuscript to PLOS ONE. After careful consideration, we feel that it has merit but does not fully meet PLOS ONE’s publication criteria as it currently stands. Therefore, we invite you to submit a revised version of the manuscript that addresses the points raised during the review process.

We look forward to receiving your revised manuscript.

Kind regards,

Dr. Chinmay Laxmeshwar

Academic Editor

PLOS ONE

Additional Editor Comments:

I commend the authors on the hard work. However, there are still gaps in the manuscript that need to be addressed. In addition to the reviewers comments, please also consider the comments below. Also, please check the language and use formal academic language.

1. Lines 50-51: Please reconsider the use of, “to reduce the problem of HIV”

2. Line 52: “Ethiopians are currently” is not the correct tense

3. Line 54: Please rephrase “infected children” this phrase. Look at the UNAIDS guidance on terminology at https://www.unaids.org/en/resources/documents/2015/2015_terminology_guidelines

4. Lines 60-61: The UNAIDS goals are now 95-95-95 and not 90-90-90

5. Lines 64-65: Please specify which WHO treatment failure criteria – clinical, immunological, or virological

6. Line 156: How many charts were excluded and why?

7. After reading “This showed that almost an equivalent proportion of male and female sample as taken for the study”, it feels like you have purposively taken this proportion.

8. Table 2: In Status last visit, please clarify what are “drop” and “lost”

9. Hazard ratios are not interpreted as “times higher/lower risk”. Look at PMID: 15273082

Reviewers' comments:

Reviewer's Responses to Questions

**Comments to the Author**

1. If the authors have adequately addressed your comments raised in a previous round of review and you feel that this manuscript is now acceptable for publication, you may indicate that here to bypass the “Comments to the Author” section, enter your conflict of interest statement in the “Confidential to Editor” section, and submit your "Accept" recommendation.

Reviewer #2: All comments have been addressed

Reviewer #4: (No Response)

2. Is the manuscript technically sound, and do the data support the conclusions?

Reviewer #2: (No Response)

Reviewer #4: Partly

3. Has the statistical analysis been performed appropriately and rigorously? 

Reviewer #2: (No Response)

Reviewer #4: Yes

4. Have the authors made all data underlying the findings in their manuscript fully available?

Reviewer #2: (No Response)

Reviewer #4: Yes

5. Is the manuscript presented in an intelligible fashion and written in standard English?

Reviewer #2: (No Response)

Reviewer #4: Yes

6. Review Comments to the Author

Reviewer #2: (No Response)

Reviewer #4: (No Response)

7. PLOS authors have the option to publish the peer review history of their article (what does this mean?). If published, this will include your full peer review and any attached files.

Reviewer #2: **Yes: **Parvati Nair

Reviewer #4: No

---

## [Author Response · Author response to Decision Letter 2]

13 Apr 2023

Response to Reviewers

Title: Incidence and Predictors of Antiretroviral Treatment Failure among Children in Public Health Facilities of Kolfe Keranyo Sub-City, Addis Ababa, Ethiopia: Institution-based retrospective cohort study

Authors:

Meseret Misasew (mesimisasew@gmail.com)

Takele Menna (takele.menna@sphmmc.edu.et)

Eyoel Berhanu (eyoel.berhanu@sphmmc.edu.et)

Daniel Angassa (danielssa25@gmail.com)

Yesunesh Teshome (yesunesh.teshome@yahoo.com)

Version: 3 Date: 12 April 2023

Response to reviewers’ comments and suggestions

We would like to thank the reviewers for taking the time to read our paper and forward their valuable comments and suggestions which in turn helped us to make a significant improvement to our manuscript. We have attempted to address all the reviewers' comments and suggestions and carefully took corrective measures. Additionally, we have included two additional authors, Takele Menna and Eyoel Berhanu, full information including affiliation is included in the revised manuscript. These changes can be found highlighted in light blue color in the title page and Authors’ contribution section of the revised manuscript, lines 5, 8, 9, 10, 11, 297, 300, 302, 308, 311, and 313.

Furthermore, the point-by-point response to the reviewers’ comments is presented below. Our responses are written in blue font color. 

Academic Editor:

1. Lines 50-51: Please reconsider the use of, “to reduce the problem of HIV”

Authors’ response: Thank you for your important suggestion. We have rephrased the mentioned phrase as suggested. This change can be found highlighted in the introduction section of the revised manuscript, line 52.

2. Line 52: “Ethiopians are currently” is not the correct tense

Authors’ response: Thank you for your valuable comment. We have corrected the tense of the mentioned sentence as suggested. This change can be found highlighted in the introduction section of the revised manuscript, line 54.

3. Line 54: Please rephrase “infected children” this phrase. Look at the UNAIDS guidance on terminology at https://www.unaids.org/en/resources/documents/2015/2015_terminology_guidelines

Authors’ response: Thank you for your valuable comment. We have rephrased the mentioned phrase as suggested. This change can be found highlighted in the introduction section of the revised manuscript, lines 56 and 57.

4. Lines 60-61: The UNAIDS goals are now 95-95-95 and not 90-90-90

Authors’ response: Thank you for your important update. We have corrected the UNAIDS target as suggested. This change can be found highlighted in the introduction section of the revised manuscript, line 62.

5. Lines 64-65: Please specify which WHO treatment failure criteria – clinical, immunological, or virological

Authors’ response: Thank you for your important suggestion. We have specified the mentioned sentence as suggested. This change can be found highlighted in the introduction section of the revised manuscript, line 67.

6. Line 156: How many charts were excluded and why?

Authors’ response: Thank you for your important question. No charts were excluded and all of the children who were enrolled in first-line Antiretroviral treatment during the study period were included.

7. After reading “This showed that almost an equivalent proportion of male and female sample as taken for the study”, it feels like you have purposively taken this proportion.

Authors’ response: Thank you for raising your vital concern. We inadvertently obtained an almost proportional sample size, without intending to do so.

8. Table 2: In Status last visit, please clarify what are “drop” and “lost”

Authors’ response: Thank you for your important suggestion. We have provided a footnote on Table 2 as suggested. This change can be found highlighted in the results section of the revised manuscript, lines 177-179.

9. Hazard ratios are not interpreted as “times higher/lower risk”. Look at PMID: 15273082

Authors’ response: Thank you for your important suggestion. As our study does not involve a clinical trial, we have interpreted the hazard ratios in a manner that aligns with the interpretation methods of most of similar studies including our references. 

Reviewer:

Major Observations:

1. The authors may reconsider the outcome variable: Incidence/Incidence density of Treatment

failure. Since the cohort of children enrolled in first-line ART were considered for the study,

it could be presented as a proportion of group who developed treatment failure. 

Authors’ response: Thank you for your valuable suggestion. We have ensured consistency throughout the manuscript by designating 'incidence' as the outcome variable in all sections. 

2. The Incidence and Incidence density are used interchangeably in the manuscript (Abstract,

Methods, Result section)

Authors’ response: Thank you again for your valuable comment. We have amended and ensured consistency throughout the manuscript by using the word 'incidence' instead of ‘incidence density’ in all sections. This change can be found highlighted in the abstract section of the revised manuscript, line 33.

3. The units for Incidence are also used interchangeably between person-months and person

years (Line 141: person-months; Line 182: person-years)

Authors’ response: Thank you again for your valuable comment. We have amended and ensured consistency throughout the manuscript by using the unit 'person-months' instead of ‘person-years’ in all sections. This change can be found highlighted in the results section of the revised manuscript, line 186.

4. Line 279 (“Enhancing parents or caregivers and health care providers support to children on

antiretroviral treatment adherence is essential for achieving the targeted viral suppression.”): It will be helpful for policy makers and readers if authors add specific recommendation, as per the context and study findings. The recommendation seems general. Authors may suggest ways to enhance the support for children on ART.

Authors’ response: Thank you again for your valuable comment. We have amended the mentioned sentence as suggested. This change can be found highlighted in the conclusions section of the revised manuscript, lines 283-285.

Minor Observations:

1. Line 31: Rate is mentioned with a %. Please revise

Authors’ response: Thank you for your important suggestion. We have amended the mentioned word accordingly. This change can be found highlighted in the abstract section of the revised manuscript, line 33.

2. Line 93 (“Additionally, after considering a 10% non-response rate for incomplete medical

records, the final sample size was 250.”): Non-response rate are usually related to survey

methods; authors should consider revising the phrase.

Authors’ response: Thank you again for your important suggestion. We have amended the mentioned sentence as suggested. This change can be found highlighted in the methods section of the revised manuscript, line 95.

3. Line 95 (“Facilities with more than ten pediatric ART clients were selected.”): Authors must

add what time period was considered for the category, 10 patients in a month/year?.

Authors’ response: Thank you again for your valuable comment. We amended the mentioned sentence as suggested. This change can be found highlighted in the methods section of the revised manuscript, line 97.

4. Line 156 (”A total of 250 medical charts were included in this study.”): Authors must explain

if each medical chart for one child or more than one medical chart were included for one

child in the study.

Authors’ response: Thank you again for your important suggestion. We amended the mentioned sentence as suggested. This change can be found highlighted in the results section of the revised manuscript, line 157.

5. Line 168 (”More than half (63.6%)”): Authors must add the numbers also in the results section..

Authors’ response: Thank you again for your suggestion. We tried to simplify the results description to ensure its easily understandable for readers. 

6. Results section: The results must be described in the text, as per occurrence in the table.

Authors’ response: Thank you again for your important suggestion. We tried to make the results description in accordance with the corresponding tables. 

7. Line 177: Multiple phrases are used for study participants: children on ART/ medical charts/study subjects. Authors must use uniform description across the manuscript.

Authors’ response: Thank you again for your important comment. We have amended and ensured consistency of the mentioned words throughout the entire manuscript.

8. Table 2: Authors may add description of category: Status last visit. Lack of clarity between

the sub-category: Drop and Lost. The descriptions can be added in the Methods section, and

as a footnote to the table. 

Authors’ response: Thank you again for your valuable comment. We have accepted your comment and add the footnote on Table 2. This change can be found highlighted in the results section of the revised manuscript, lines 177-179.

Kind regards,

Meseret Misasew

Corresponding author

---

## [Decision Letter · Decision Letter 3]

22 Jun 2023

Incidence and Predictors of Antiretroviral Treatment Failure among Children in Public Health Facilities of Kolfe Keranyo Sub-City, Addis Ababa, Ethiopia: Institution-based retrospective cohort study

PONE-D-22-08487R3

Dear Dr. Misasew,

We’re pleased to inform you that your manuscript has been judged scientifically suitable for publication and will be formally accepted for publication once it meets all outstanding technical requirements.

Kind regards,

Chinmay Laxmeshwar

Academic Editor

PLOS ONE

Additional Editor Comments (optional):

Reviewers' comments:

Reviewer's Responses to Questions

**Comments to the Author**

1. If the authors have adequately addressed your comments raised in a previous round of review and you feel that this manuscript is now acceptable for publication, you may indicate that here to bypass the “Comments to the Author” section, enter your conflict of interest statement in the “Confidential to Editor” section, and submit your "Accept" recommendation.

Reviewer #2: All comments have been addressed

Reviewer #4: (No Response)

2. Is the manuscript technically sound, and do the data support the conclusions?

Reviewer #2: Yes

Reviewer #4: Yes

3. Has the statistical analysis been performed appropriately and rigorously? 

Reviewer #2: Yes

Reviewer #4: Yes

4. Have the authors made all data underlying the findings in their manuscript fully available?

Reviewer #2: Yes

Reviewer #4: Yes

5. Is the manuscript presented in an intelligible fashion and written in standard English?

Reviewer #2: Yes

Reviewer #4: Yes

6. Review Comments to the Author

Reviewer #2: (No Response)

Reviewer #4: The authors have incorporated most of the suggested changes. However, the following points could be reconsidered by the authors:

1. Line 32_"The overall treatment failure rate within the follow-up period was 17.2%": Authors may reconsider mentioning rate with a percentage. It should be mentioned as a proportion.

2. Line 157_"A total of 250 medical charts were included in this study.": Though it is mentioned that the change has been incorporated, however the statement seems the same as before. The authors should revise the given statement.

3. Line 181_"It was found that a total of 43 subjects (17.2%) have treatment failure": Authors should reconsider the tense of the statement, in line with other statements in the paragraph.

END

7. PLOS authors have the option to publish the peer review history of their article (what does this mean?). If published, this will include your full peer review and any attached files.

Reviewer #2: No

Reviewer #4: No

---

## [Editor Report · Acceptance letter]

11 Aug 2023

PONE-D-22-08487R3 

Incidence and Predictors of Antiretroviral Treatment Failure among Children in Public Health Facilities of Kolfe Keranyo Sub-City, Addis Ababa, Ethiopia: Institution-based retrospective cohort study 

Dear Dr. Misasew:

I'm pleased to inform you that your manuscript has been deemed suitable for publication in PLOS ONE. Congratulations! Your manuscript is now with our production department. 

Kind regards, 

on behalf of

Dr. Chinmay Laxmeshwar 

Academic Editor

PLOS ONE